# The Influence of Proprioceptive Training with the Use of Virtual Reality on Postural Stability of Workers Working at Height

**DOI:** 10.3390/s20133731

**Published:** 2020-07-03

**Authors:** Magdalena Cyma-Wejchenig, Jacek Tarnas, Katarzyna Marciniak, Rafał Stemplewski

**Affiliations:** Poznan University of Physical Education, 61-871 Poznan, Poland; tarnas@awf.poznan.pl (J.T.); katarzyna.anna.m@gmail.com (K.M.); stemplewski@awf.poznan.pl (R.S.)

**Keywords:** postural stability, virtual reality, proprioceptive training, at‒height workers

## Abstract

The aim of the study was to assess the impact of proprioceptive training with the use of virtual reality (VR) on the level of postural stability of high–altitude workers. Twenty-one men working at height were randomly assigned to the experimental group (EG) with training (*n* = 10) and control group (CG) without training (*n* = 11). Path length of the displacement of the center of pressure (COP) signal and its components in the anteroposterior and medial–lateral directions were measured with use of an AccuGaitTM force plate before and after intervention (6 weeks, 2 sessions × 30 min a week). Tests were performed at two different platform heights, with or without eyes open and with or without a dual task. Two–way ANOVA revealed statistically significant interaction effects for low–high threat, eyes open-eyes closed, and single task-dual task. Post-training values of average COP length were significantly lower in the EG than before training for all analyzed parameters. Based on these results, it can be concluded that the use of proprioceptive training with use of VR can support, or even replace, traditional methods of balance training.

## 1. Introduction

Construction industries have the highest statistics of injuries and fatalities and working at height involves a particularly high risk [1]. The risk is associated with being at high altitudes, often combined with difficult weather conditions, e.g., strong winds, rain and snow [2]. Over 10% of all fatal accidents in the construction industry are a consequence of falling from elevated surfaces [3]. According to the Occupational Safety & Health Administration [4] any work where the level difference between the workplace and the floor creates the threat of falling from one level to a lower level is working at height, for example: on scaffolding, ladders or other elevations. For people working from one meter upwards, examinations of the balance system are recommended [5]. Loss of balance, slips and trips are the most common causes of accidents [6,7].

In order to prevent accidents among high-altitude workers, researchers are focusing on indicating factors that affect instability and loss of balance. Maintaining stability and balance of posture is difficult due to continual changes in the vertical projection of the center of mass (COM) on the base of support, which may be affected by environmental and internal factors [8]. The stability of human posture is associated with three main sensory senses: the visual, vestibular and somatosensory systems [9]. These senses are closely related to correctly interpreting reality and responding to it appropriately. Such sensory integration is associated with the function of the central nervous system, as well as a neurological process that organizes sensations flowing from the body and the environment in such a way that they can be used for purposeful action [10,11].

In cases where one of the sensory systems is at risk, or the sensory information is inaccurate, postural stability (PS) and balance may be disturbed, especially when the changes affect the visual system [12]. Hsiao and Simeonov [13] noticed in their studies that moving visual scenes and depth perception are among the key elements provoking instability and negatively affecting balance.

It was also found that lowering PS level is affected by performing tasks at height. Stability disorders during performance of tasks at height are enhanced by visual stimuli, which cause additional anxiety because task performance seems to be dangerous [14,15,16,17]. Another factor affecting PS may be performing a dual task or switching between tasks. The effectiveness of posture control while performing a dual task may decrease compared to performing a single task [18]. In addition, it has been shown that it is very important to freely divide attention to prevent loss of balance, and falls as a consequence, when performing a double task [19].

The ability of high-altitude employees to identify and assess risk is acquired through training and experience. It is one of the key factors determining their behavior, and thus their safety. Training and drills in this area are a very important element in protecting employees from falling [20,21,22]. In education and training programs in construction engineering, virtual reality (VR) technologies have quickly gained recognition [21]. For example, Goedert et al. [23] developed a virtual interactive educational platform to provide safety training. The training was based on games through the use of simulation and modeling. Overall, the project proved to be both effective and engaging for employees.

Donath, Rossler and Faude [24] demonstrated in their studies thatnew technologies such as VR may also be used during training. Thanks to VR technologies, it is possible to train and strengthen individual body parts more effectively and easily adapt the exercises to individual abilities and needs [8]. According to research, training using VR positively affects the improvement of PS [25]. VR has also been integrated with applied science, e.g., these technologies have been applied in architecture and design visualization, medicine and construction health and safety, enabling further improvements in the efficiency of education and training in the construction industry, in particular among high-altitude workers [21].

Recently, training platforms facilitating proprioceptive exercises with the use of VR, which recreate a natural sense of instability because the body is forced to do more work, have become popular training devices [26]. Through these, the muscles can be trained, reaction ability stimulated, and body balance shaped [27,28].

This type of training may bring about additional effects compared to those achieved during standard equivalent exercises, which can be predicted on the basis of studies on the sick and the elderly, as well as the healthy and young [28,29,30]. Furthermore, VR simulation can be used as a comprehensive system that integrates elements necessary for active learning for a group of high-altitude workers [31]. For example, Wang et al. [32] analyzed the effectiveness of using serious games in 4D technology (3D + time) in training in the field of occupational health and safety in construction. It has been noted that VR can increase users’ involvement and affect their ability to detect Occupational Safety and Health risks. Similar conclusions were presented by Strobach, Frensch and Schubert [33] who stated that the practice of video games improves executive control skills while performing a double task.

On the other hand, there are numerous studies showing that the benefits of virtual reality are often surpassed by limitations due to cybersickness. Cybersickness is similar in symptoms to motion sickness and can result in nausea, headaches and dizziness [34]. For example, Nalivaikoa et al. [35] analyzed the influence of how cybersickness, provoked by a head-mounted display, affects cutaneous vascular tone, heart rate and reaction time. It has been noted that Cybersickness evokes nausea, increases body temperature and extends reaction time, raising obvious concerns regarding the safety of this technology [35]. Therefore first, it should be checked if people participating in VR training do not report similar ailments.

To our best knowledge, no research has been conducted regarding the effects of proprioceptive training on a balance platform using VR for individual PS parameters in high-altitude workers. However, based on the research of the elderly, the sick and those with neurological disorders, as well as healthy people [36,37,38], it can be assumed that this type of training can also produce positive effects in high-altitude workers. For example, Amritha et al. [39] studied the effect of using a balance platform that provides static and dynamic balance training through interactive VR games for those with balance disorders. Studies have shown that balance training using a balance platform significantly improves PS levels and has a positive effect on daily activities.

The aim of the work was to assess the impact of proprioceptive training using VR on the PS level of employees working at height. We assumed the general hypothesis that proprioceptive training with VR using a balance platform has a positive effect on the level of PS. Moreover, we assumed that proprioceptive training with VR would contribute to increased stability in the case of (1) a standard stability test with eyes open, (2) reduction of visual stimuli, (3) changing the height of the test plane and (4) introducing an additional cognitive task.

## 2. Materials and Methods

### 2.1. Characteristics of the Research Group

The study involved 24 healthy men working at height between the ages of 22–47. High-altitude workers (HW) were randomly assigned (Excel software) to the study and divided into two groups:Experimental group (EG)—HW training on a balance platform using VR: initial *n* = 12;Control group (CG)—HW not training on a balance platform using VR: initial *n* = 12.

During the study, 2 men in the EG withdrew from the study due to injury (injury was not related to the experiment) and 1 man in the CG withdrew from the study without giving a reason. As a result, 21 men in both groups participated in the study (Figure 1). The eligibility criteria were as follows: at least one year of experience in working at height, minimum age of 20, full mobility, verbal contact skills enabling informed and logical answers.

All men participating in the experiment were informed in detail about the study procedures and gave their written consent for the experimental procedure. Participation in the experiment was voluntary. None of the subjects had experience in training on a balance platform using VR. The basic characteristics of both groups are presented in Table 1. The study was approved by the Bioethics Committee of the Poznan University of Medical Sciences (decision no. 1111/16) and was in line with the Helsinki Declaration [40].

### 2.2. Experimental Procedures

#### 2.2.1. General Course of the Experiment

A general review of the experiment is shown in Figure 2. Intervention with the use of VR on the balancing platform was introduced in the subjects. The level of postural stability was examined before and after the intervention. The research was carried out at the Poznan University of Physical Education in Poland, at the Department of Physical Activity and Health Promotion Science.

#### 2.2.2. Initial Measurements

Before the experiment, measurements of somatic characteristics were carried out, i.e., height and weight measurement and BMI (Body Mass Index). In addition, patients were interviewed and their overall health and well-being assessed. The level of physical activity was assessed using Caltrac (Muscle Dynamics, Inc., Tarrance, CA) accelerometer reports, and the results were based on weekly measurement of the energy expenditure associated with physical activity [41,42]. Subjects wore the Caltrac for 7 days. The total result in kilocalories was divided by the number of days and body weight (PA–EE) to obtain normalized values.

#### 2.2.3. Intervention

The experimental group participated in proprioceptive training on a balance platform for 6 weeks for 30 min per session. The trainings took place twice a week. The total number of training sessions was 12, and 100% participation in the training sessions was required. A balance platform was used for the training, Sigma (ACX.rehab element), which is a modern device for training proprioception using VR (Sigma balance platform, Prod. AC International). It was equipped with an independent system for assessing the swing angle of the platform using a gyroscope. The sampling frequency was 40 Hz and the delay of 25 ms. The sensor registered every change in the position of the platform, converting these changes into an appropriate output signal and transmitting data in real time to a computer with the software. This transmission was done wirelessly using a Bluetooth sensor.

During training, audio-visual feedback for participants was used, i.e., biofeedback in the form of video games facilitating exercises through play. The games were simple and involved moving objects like a fish, plane, car and balls. For instance, in the fish game, the subject mainly practiced movement precision. The subject had the task of moving a blue circle to protect the fish from a source of sparks, which was safe when it was in the center of the circle (Figure 3). The game made it possible to measure and practice individual skills in implementing specific movement patterns while maintaining a set speed and range. The goal of the exercise was to repeat movements in 3D space, planned movement training, precision of movements and improving muscle strength. Alternatively, in the sample plane game, the subject practiced functional movements. The subject’s task was to fly a plane through circles. The closer to the circle they flew, the more points they scored (Figure 3). The subject was thus able to practice focus, perceptiveness, precision of movements, predicting the trajectory of moving objects in 3D space, as well as balance during the game.

After each training, the subject could see the result obtained in a given game. After each game, the program analyzed the accuracy, number of points and precision of moves and also showed the progress that was achieved between individual trainings (Figure 4).

Exercises on the platform began with the simplest ones, and then the subjects proceeded to more complex and complicated exercises. The guidelines for the experimental group were as follows: the subject stood motionless in the center of the platform with feet spaced at hip width so that the feet were on opposite sides of the disk, parallel to each other. The platform was placed 2 m in front of the monitor stand where the games were broadcast. The subject was asked to perform a task according to instructions in the games. During the intervention, the subject balanced their body standing on the platform and carried out the exercises using their body and limbs (Figure 5). Exercises on the balance platform took place in an area where there was free space so that in the event of falling off the disk there was no risk of injury. During the tests, the person being examined was supported by the researcher who, in addition to physical assurance, also verbally controlled the well-being of the examinee. As part of their participation, each subject was asked to play all 9 games every time. The game time could be set through the program and the length of a single game was set between 2–3 min so that the whole single session lasted 30 min. Respondents were also able to rest in the sitting position for 30–60 s after each exercise.

### 2.3. Measurement

#### 2.3.1. Postural Stability (PS)

The level of postural stability was measured before and after the intervention using an AccuGaitTM power plate system (Model AMTI PJB-101, AMTI, Watertown, MA, USA) with Balance Trainer software. The sampling frequency was 100 Hz.

The research procedure included posturographic tests assessing the balance of the human body on a stabilometric platform at ground level and at a height of 1 m from the ground. A single trial lasted 30 s and was carried out twice (the average of 2 trials was taken into account). A 20 s break was used between measurements due to the large number of trials. In the event of fatigue or dizziness, participants could rest in the sitting position. None of the participants took advantage of this opportunity, which is why during all tests at low or high level they stayed on the platform at the same level. A stable, one-meter-high pedestal was used to place the stabilometry platform at a height. Participants climbed the pedestal via three-step stairs. Gym mattresses were positioned around the platform to ensure safety. The subject, both on the ground and at height, performed tests with open and closed eyes and with or without a cognitive task. During the examination, the subject stood still in the center of the platform with their bare feet spaced at hip width and arms along the torso. The posturographic platform was placed 3 m in front of a white wall, and subjects were asked to look directly at the wall. The following tests were carried out in random order to avoid the learning factor:EO LT QS—Eyes Open—Low Threat (ground level)—Quiet Standing.EO LT DT—Eyes Open—Low Threat—Dual Task (the subject additionally performed a mathematical task, which consisted of counting every 3 numbers down from 200, for the period of the test registration [43])EO HT QS—Eyes Open—High Threat (1m above the ground)—Quiet Standing.EO HT DT—Eyes Open—High Threat—Dual Task.

All tasks with the eyes closed (EC) were performed in the same way as with the eyes open (EC LT QS, EC LT DT, EC HT QS, EC HT DT, respectively).

The study took into account the length of the displacement path of the center of pressure (COP) signal and its elements in the anteroposterior (AP) and medial-lateral (ML) directions. It is widely used as a PC indicator [44]. Each test was performed twice and the results were averaged. Earlier studies confirmed that averaging 2 results was sufficient to obtain the Intraclass Correlation Coefficient (ICC) reliability coefficient above 0.9 for the average COP displacement speed (constant parameter equals path length) [45].

#### 2.3.2. Statistical Analysis

The main calculations related to the assessment of the variability of dependent variables were calculated based on the ANOVA variance analysis method (test F). The analysis was applied taking into account the intergroup factor (with EG and CG levels) and the repeated factor of the time measurement (with the pre and post levels). The aim of the study was not focused on differences between conditions effects so that’s why there were counted interaction effects (group x time) for each condition separately. Similar calculations were used for all analyzed conditions related to the experiment: EO LT QS, EO LT DT, EO HT QS, EO HT DT, EC LT QS, EC LT DT, EC HT QS and EC HT DT.

The difference assessments between groups in terms of basic characteristics (age, BMI, physical activity indicators) were made using the Student’s t‒test. The minimum level of statistical significance was *p* ≤ 0.05. The study was conducted using the Statistica v. 13.0 software program (TIBCO Software Inc., Palo Alto, CA, USA).

## 3. Results

### 3.1. Effect of the Intervention on Postural Stability with Eyes Open in Conditions of the Low-High Threat during Quiet Standing or Performing a Dual Task (EO LT QS, EO HT QS, EO LT DT, EO HT DT)

In the case of quiet standing with the eyes open under low threat conditions, a significant interaction effect was found only for the sway path medial-lateral (SPML) (*F* = 5.47, *p* < 0.05, η^2^ = 0.22). In an analogical analysis for high threat conditions, a significant interaction effect was observed for the sway path (SP), the sway path anteroposterior (SPAP) and SPML (*F* = 14.28, *p* < 0.001, η^2^ = 0.43; *F* = 8.1, *p* < 0.01, η^2^ = 0.29 and *F* = 24.78, *p* < 0.000, η^2^ = 0.57, respectively). For all studied variables, a significant main time effect was also found (η^2^ between 0.25 and 0.52). In the EG, the post-training values were significantly lower than the baseline values (Table 2).

In the dual-task study conditions, a significant effect of “time x group” interaction was found for SP, SPAP and SPML when measured at ground level (*F* = 12.50, *p* < 0.01, η^2^ = 0.40; *F* = 12.38, *p* < 0.01, η^2^ = 0.39; *F* = 7.29, *p* < 0.01, η^2^ = 0.28, respectively) and at the increased height of the measuring station (*F* = 35.65, *p* < 0.0000, η^2^ = 0.65; *F* = 31.72, *p* < 0.0000, η^2^ = 0.62; *F* = 21.17, *p* < 0.0001, η^2^ = 0.52, respectively). A significant main effect (time) was recorded for SP, SPML and SPAP during measurement at ground level (*F* = 19.0, *p* < 0.000, η^2^ = 0.50; *F* = 15.29, *p* < 0.001, η^2^ = 0.44 and *F* = 20.05, *p* < 0.000, η^2^ = 0.51, respectively) and for SP when measured at the elevated altitude (*F* = 5.03, *p* < 0.05, η^2^ = 0.21). After VR training, lower values of tested variables in the EG were observed during the trial with dual–task (Table 3).

### 3.2. Effect of the Intervention on Postural Stability with Eyes Closed in Conditions of Low-High Threat during Quiet Standing or Performing a Dual Task (EC LT QS, EC HT QS, EC LT DT, EC HT DT)

A significant interaction effect was observed during quiet standing with the eyes closed for all studied parameters SP, SPAP and SPML at both low threat (*F* = 14.17, *p* < 0.001, η^2^ = 0.43; *F* = 12.42, *p* < 0.01, η^2^ = 0.39 and *F* = 8.95, *p* < 0.01, η^2^ = 0.32, respectively) and high threat (*F* = 17.24, *p* < 0.000, η^2^ = 0.47; *F* = 15.72 *p* < 0.001, η^2^ = 0.45 and *F* = 7.33, *p* < 0.01, η^2^ = 0.27, respectively) altitudes. Post-training values of average COP length were significantly lower in the EG than before training. A significant main effect (time) was found for all studied parameters except for SPML-high threat (>0.05) (Table 4).

In the case of dual-task with the eyes closed, a significant interaction effect was found for all parameters studied during measurement at ground level (*F* = 7.36, *p* < 0.01, η^2^ = 0.27; *F* = 6.15, *p* < 0.05, η^2^ = 0.24 and *F* = 20.49, *p* < 0.001, η^2^ = 0.52, respectively). At the increased height of the measuring station (high threat) during a dual task, a significant effect of the “task x group” interaction was found for SPAP (*F* = 6.56, *p* < 0.05, η^2^ = 0.25). Detailed results of the analysis of variance are presented in Table 5.

## 4. Discussion

### 4.1. Effect of Intervention on Postural Stability with Eyes Open

Balance training is necessary to maintain and improve the level of PS, especially in high-altitude workers [8,32]. In this study, we used proprioceptive training on a balance platform using VR technology to improve the PS level of high–altitude workers.

The obtained results confirmed that training significantly improved PS level both under conditions on the ground and with increased height of the measuring platform [33,46]. In quiet standing tests with visual inspection, both a decrease of the overall length of the COP movement path and its components in the forward-backward and right-left directions at both heights were obtained. The EG group achieved lower results (which may indicate a higher level of postural stability) in all analyzed parameters after training. Higher levels of PS in the EG group probably resulted from increased involvement of the visual, vestibular and somatosensory systems, as well as individual muscle groups used in tasks performed on the balance platform during VR training. Based on the results, it can be assumed that even in the case of visual-vestibular-sensory conflict, the EG group would probably maintain the correct PS level, which may reduce the percentage of accidents while working at height. The results obtained at height also suggested that training using VR eliminated the negative impact of height as well as visual stimuli on the level of postural stability.

Carozza et al. [47] used virtual reality goggles to train construction workers in risk recognition skills. The authors of the study stated that performing realistic training reduced the risk associated with performing tasks in real conditions. Positive results were reported especially in the context of work at height. There is no other research on the impact of proprioceptive training on a balance platform using VR on the PS level of high-altitude employees. However, indirect evidence can be sought in studies using interactive training in the elderly or the sick.

Schwenk et al. [29] conducted pilot studies of interactive balance training in the elderly based on visual motion feedback sensors. The authors noted that older people at risk of falling may benefit from participating in a balance training program. In addition, this training may in the future support traditional balance training or completely replace it. In turn, Srivastava at al. [30] noted that even in the chronic phase after stroke, significant improvement in balance and functional outcome can be achieved after training on a balance platform with Visual Feedback (FPVF).

In cognitive standing tests at visual control, the EG group achieved both a reduction in the overall length of the COP path and its components in the forward-backward and right-left directions at both heights. We can assume that the higher level of PS in the EG group probably resulted from an increase in cognitive involvement when performing realistic tasks in VR training [33,48]. Experience in action games results in the improvement of a wide range of perceptual skills. Players also show an improvement in other cognitive skills, in particular the ability to effectively switch between tasks [49,50]. So far, no studies on the impact of proprioceptive training on a balance platform using VR on the PS level in a dual task with the eyes open have been conducted in high-altitude workers. However, research among players has shown the positive impact of games on cognitive performance.

Colzato et al. [51] studied whether, and to what extent, experience in video games affected the performance of a cognitive task. Experienced players were faster and more accurate in monitoring and updating working memory, and also reacted faster to the signals sent compared to inexperienced players. The authors suggest that the use of games was associated with increased flexibility in updating information while performing a cognitive task. In turn, Strobach, Frensch and Schubert [33] noticed that when performing a task that requires split attention, action-game players had an excellent ability to perform two tasks at the same time and to switch efficiently between tasks.

### 4.2. Effect of the Intervention on Postural Stability with Eyes Open

When visual control is restricted, mainly proprioceptive information is used. In this study, during the trial, we turned off the visual effect to check whether proprioceptive training on a balance platform using VR would have a positive effect on the level of PS of high-altitude workers.

The EG group in quiet standing tests without visual control obtained both a reduction in the length of the COP path as well as both its components towards AP and ML. Significant differences were also found in all parameters of the COP path length in the test at the elevated height of the measuring station. This indicated that training had a positive effect on the level of PS in the EG regardless of the altitude.

Based on the result, we can assume that the EG group relied more on proprioception, and less on visual input to maintain balance, than the CG, which was associated with increased ability to maintain balance. Research has shown that this type of ability can be successfully developed [52]. It is likely that the EG group acquired such skills during proprioceptive training on a balance platform using VR. Hutt and Redding [52] studied the impact of specific training on balance control among ballet dancers. They stated that a program based on reducing visual stimuli increased the level of balance. Other studies have found that training with the Nintendo Wii Fit console improved balance and gait in those with Parkinson’s disease [53].

Positive training effects were also obtained in tests at ground level with the eyes closed during a dual task. The EG group achieved lower results in all analyzed parameters. Presumably, the better PS in the EG group resulted from the combination of proprioceptive training with the daily training of individual muscle groups used in the professional tasks of high-altitude workers, as well as the conditions and nature of the work [2,11,51].

In both groups, no significant changes were observed in the PS level with an increase in perceived difficulty in maintaining balance due to height and task performance. Despite the fact that the training was performed only with the eyes open, the measurement results with the eyes closed were close to statistically significant. The result may have been affected by the difficulty of the task being performed, as well as insufficient ability to rely on proprioception in the absence of visual control. Perhaps continuing training would also result in significant differences in the test with the eyes closed at height.

Van Diest et al. [54] studied the impact of 6 weeks of training using games on the level of balance, where participants trained at home. The training led to a reduction in postural swaying after 6 weeks of participation in healthy older people in all analyzed tests.

Research on the impact of video games on cognitive performance suggests that some types of video games may increase attention, memory capacity, working memory, and performance of dual tasks. This demonstrates the potential of usefulness of these commercial games for practical applications in the real world, such as rehabilitation or training of skills related to work at height [55]. The above results have been confirmed by the results of the research obtained in this article.

Finally, it should be mentioned that this work was limited. It only included VR technologies related to games on a balance platform using proprioceptive training. In the future, VR glasses could be implemented to deepen the immersion and learning processes through experiences in VR. For example, Angelov [56] found out that the usage of the balance training program while using VR glasses (with increased difficulty level conditions) led to balance stability improvement. Similar conclusions were presented by Duque et al. [57] who studied the effect of balance training using a new VR system (the Balance Rehabilitation Unit) in older fallers. It has been showed that training is an effective and well-accepted intervention to improve balance, increase confidence and prevent falls in the elderly.

Secondly, the CG did not take any training exercise, so it’s difficult to separate between two factors acting here, namely, proprioceptive training and VR. In future studies, the control group should participate in training on the platform without VR feedback to separate the influence of these two factors.

The results of the study are expected to be a useful source of information for future research or practice in implementing VR in the field of education and training among high-altitude workers to prevent accidents at height and related injuries, and even death.

## 5. Conclusions

We provided evidence that proprioceptive training on a balance platform using VR improved the PS level in all analyzed parameters of low-high threat, eyes open-eyes closed and single task–dual task. Firstly, the proprioceptive training on a balance platform using VR turned out to be an appropriate element for direct training of the visual-vestibular-sensory system in order for high-altitude employees to learn an adequate response to threats without exposing them to danger. Secondly, proprioceptive training on a balance platform using VR likely resulted in greater cognitive involvement and could also increase the effectiveness of switching between tasks. Thirdly, we can assume that the EG group, after proprioceptive training with the use of VR, relied more on proprioception, and less on visual control, to maintain balance. These facts can be used to teach high-altitude workers how to automatically respond to a threat and control safety in their work environment.

Overall, training on the platform can have a number of positive effects that seem to be difficult to achieve during standard balance exercises. It is possible that the improvement of the PS level of high-altitude workers under the influence of proprioceptive training with the use of VR will positively affect quality and safety at work and prevent falls and injuries in the future.

## Figures and Tables

**Figure 1 sensors-20-03731-f001:**
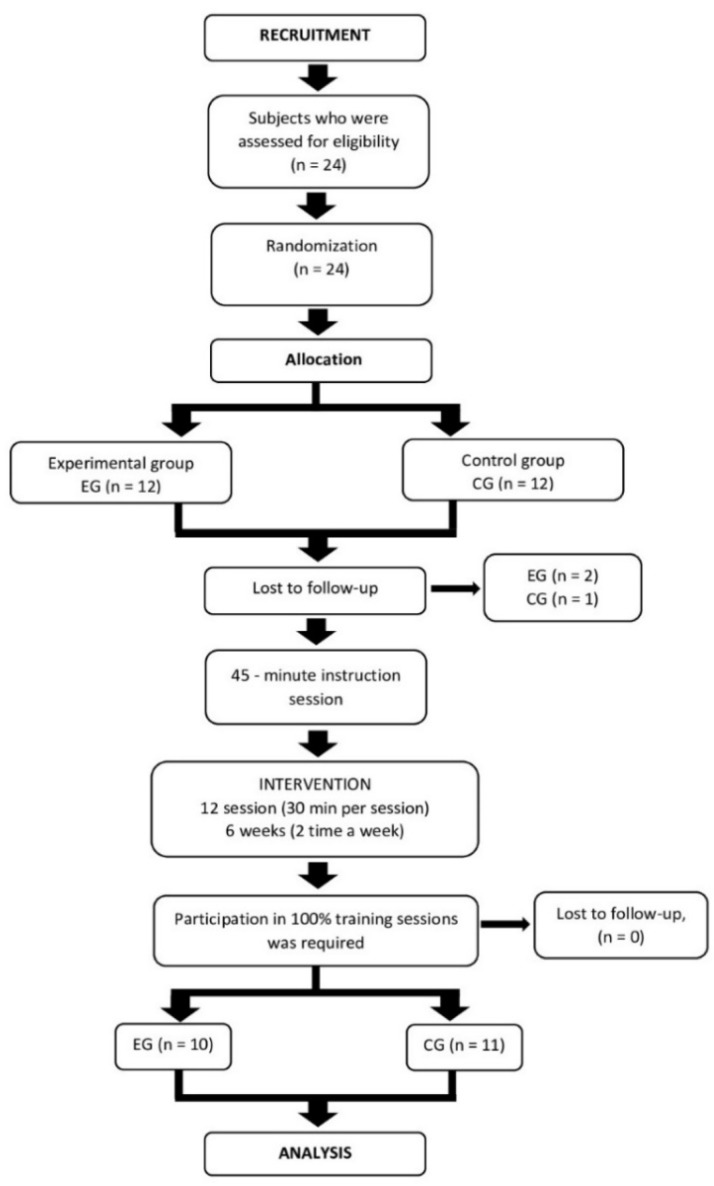
Flowchart of the study participants.

**Figure 2 sensors-20-03731-f002:**
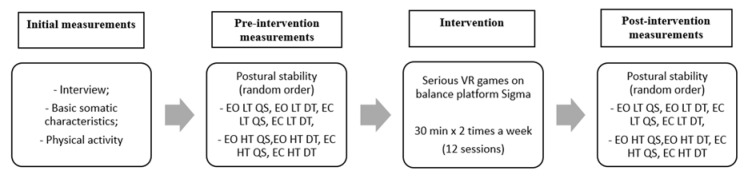
General overview of the experiment. EO—Eyes Open, EC—Eyes Open, LT—Low Threat, HT—High Threat, QS—Quiet Standing, DT—Dual Task.

**Figure 3 sensors-20-03731-f003:**
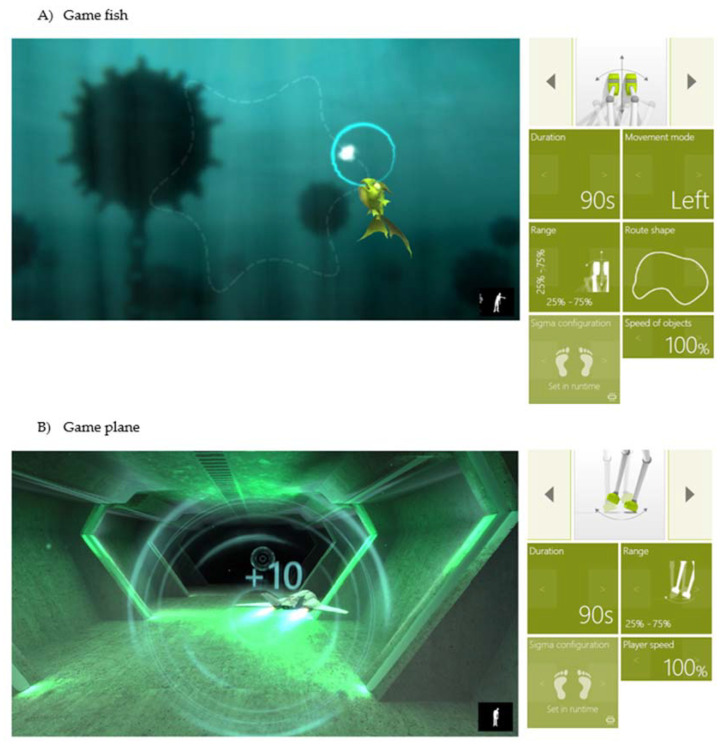
Sample settings for games (**A**) fish and (**B**) plane.

**Figure 4 sensors-20-03731-f004:**
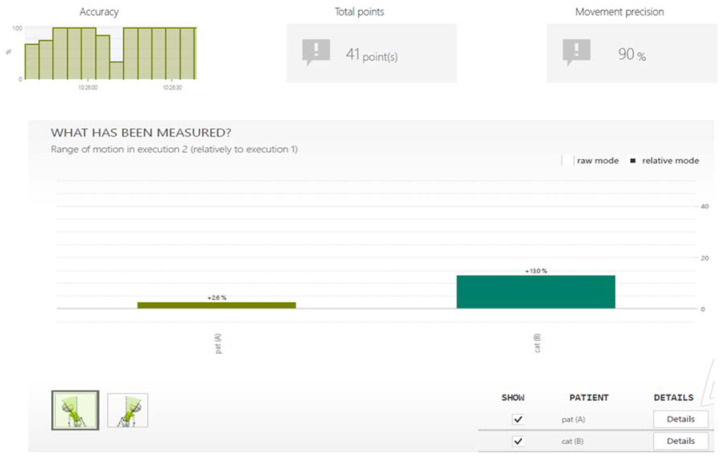
Sample analysis of results for the game fish.

**Figure 5 sensors-20-03731-f005:**
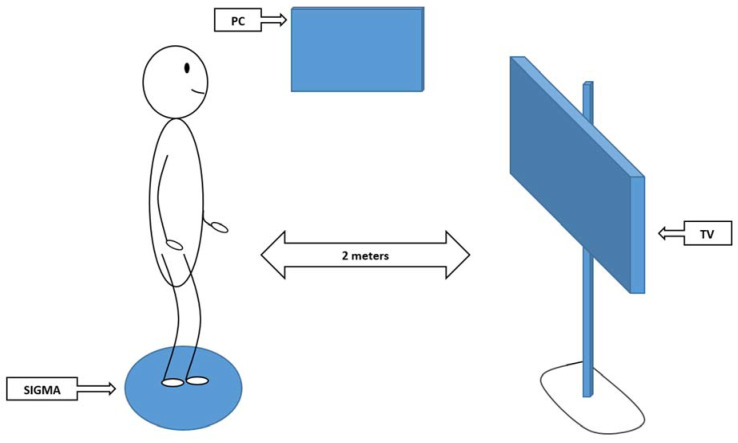
Correct position during a training session on SIGMA balance platform.

**Table 1 sensors-20-03731-t001:** Average values, standard deviations and differences between groups for the general characteristics of the participants and physical activity before the start of the experiment.

Variable	M (sd) EG	M (sd) CG	t df = 19	*p*
Age [years]	34.00(8.04)	37.27(7.87)	−0.94	0.36
Body height (m)	1.81(0.04)	1.81(0.06)	−0.37	0.72
Body weight (kg)	88.52(9.64)	88.85(13.14)	−0.06	0.95
BMI [kg/m^2^]	27.13(2.32)	26.97(3.40)	0.12	0.90
PA-E [cals/day/kg]	9.31(9.25)	11.63(9.14)	−0.57	0.57

Note. EG—experimental group; CG—control group; BMI—body mass index; PA-EE—physical activity-energy expenditure.

**Table 2 sensors-20-03731-t002:** Mean and standard deviation values of postural stability for quiet standing measures with eyes open on low and high threat in the experimental and the control groups before and after intervention and results of two-way ANOVA.

	Pre		Post							
Variable	M (sd) EG	M (sd) CG	M (sd) EG	M (sd) CG	Interaction *F*(*p*)	η^2^	Group *F*(*p*)	η^2^	Time *F*(*p*)	η^2^
Low threat										
SP-[mm]	193.1(44.32)	189.4(21.87)	166.2(36.27)	179.7(25.98)	4.02(>0.05)	0.17	0.11(>0.05)	0.01	20.47(<0.001)	0.52
SPAP-[mm]	153.7(36.55)	148.1(21.70)	131.1(28.65)	138.4(23.29)	2.66(>0.05)	0.12	0.005(>0.05)	0.00	16.82(<0.001)	0.47
SPML-[mm]	87.1(21.58)	87.9(17.38)	76.2(17.16)	85.7(18.19)	5.47(<0.05)	0.22	0.38(>0.05)	0.02	12.35(<0.01)	0.39
High threat										
SP-[mm]	210.0(48.01)	218.1(40.82)	172.7(49.91)	220.5(38.51)	14.28(<0.001)	0.43	2.25(>0.05)	0.11	11.09(<0.01)	0.37
SPAP-[mm]	169.7(40.0)	176.3(32.35)	137.1(39.53)	175.6(27.59)	8.01(<0.01)	0.29	2.51(>0.05)	0.12	8.75(<0.01)	0.31
SPML-[mm]	90.5(24.34)	93.8(27.09)	76.1(21.73)	98.(30.63)	24.78(<0.000)	0.57	11.28(>0.05)	0.06	6.52(<0.01)	0.25

Note. HW—height workers; OW—office workers; SP—Sway Path; ML—medio-lateral; AP—antero-posterior; Low threat—measurement at ground level; High threat—measurement at increased height of the measuring platform.

**Table 3 sensors-20-03731-t003:** Mean and standard deviation values of postural stability for dual-task with eyes open on low and high threat in the experimental and the control groups before and after intervention and results of two-way ANOVA.

	Pre		Post							
Variable	M (sd) EG	M (sd) CG	M (sd) EG	M (sd) CG	Interaction *F*(*p*)	η^2^	Group *F*(*p*)	η^2^	Time *F*(*p*)	η^2^
Low threat										
SP-[mm]	201.4(35.83)	211.9(47.06)	156.5(36.72)	207.3(51.90)	12.50(<.001)	0.40	2.83(>0.05)	0.13	19.0(<0.000)	0.50
SPAP-[mm]	163.0(31.89)	172.8(42.08)	122.5(29.22)	170.6(48.76)	12.38(<0.01)	0.39	3.18(>0.05)	0.14	15.29(<0.001)	0.44
SPML-[mm]	87.2(15.02)	89.5(24.73)	71.9(18.69)	85.7(21.43)	7.29(<0.01)	0.28	0.87(>0.05)	0.04	20.05(<0.000)	0.51
High threat										
SP-[mm]	210.6(33.61)	217.5(35.95)	177.5(38.32)	232.5(42.85)	35.65(<0.0000)	0.65	3.7(>0.05)	0.16	5.03(<0.05)	0.21
SPAP-[mm]	172.9(28.58)	177.8(29.21)	142.7(28.16)	191.8(33.85)	31.72(<0.0000)	0.62	4.63(<0.05)	0.19	4.29(>0.05)	0.18
SPML-[mm]	87.8(15.47)	91.5(20.39)	77.7(17.90)	95.6(23.42)	21.17(<0.0001)	0.52	1.63(>0.05)	0.07	3.76(>0.05)	0.16

Note. HW—height workers; OW—office workers; SP—Sway Path; ML—medio-lateral; AP—antero-posterior; Low threat—measurement at ground level; High threat—measurement at increased height of the measuring platform.

**Table 4 sensors-20-03731-t004:** Mean and standard deviation values of postural stability for quiet standing with eyes closed on low and high threat in the experimental and the control groups before and after intervention and results of two-way ANOVA.

	Pre		Post							
Variable	M (sd) EG	M (sd) CG	M (sd) EG	M (sd) CG	Interaction *F*(*p*)	η^2^	Group *F*(*p*)	η^2^	Time *F*(*p*)	η^2^
Low threat										
SP-[mm]	293.2(81.55)	272.2(51.41)	224.4(55.28)	270.5(48.19)	14.17(<0.001)	0.43	0.26(>0.05)	0.01	15.71(<0.001)	0.45
SPAP-[mm]	254.8(72.31)	234.0 (54.31)	195.2(48.29)	235.0(63.04)	12.42(<0.01)	0.39	0.14(>0.05)	0.01	11.62(<0.01)	0.38
SPML-[mm]	101.8(30.98)	105.2(31.39)	84.2(22.98)	107.6(34.11)	8.95(<0.01)	0.32	1.1(>0.05)	0.55	5.16(<0.05)	0.21
High threat										
SP-[mm]	279.4(70.73)	283.3(56.27)	212.1(68.06)	296.4(65.15)	17.24(<0.000)	0.47	2.72(>0.05)	0.12	7.86(<0.01)	0.29
SPAP-[mm]	238.2(64.39)	241.2(51.98)	172.4(61.38)	252.5(61.96)	15.72(<0.001)	0.45	2.92(>0.05)	0.13	7.9(<0.01)	0.29
SPML-[mm]	103.4(28.79)	104.6(23.30)	90.3(27.59)	109.6(25.18)	7.33(<0.01)	0.27	0.88(>0.05)	0.04	1.44(>0.05)	0.07

Note. HW—height workers; OW—office workers; SP—Sway Path; ML—medio-lateral; AP—antero-posterior; Low threat—measurement at ground level; High threat—measurement at increased height of the measuring platform.

**Table 5 sensors-20-03731-t005:** Mean and standard deviation values of postural stability for dual-task with eyes closed on low and high threat in the experimental and the control groups before and after intervention and results of two-way ANOVA.

	Pre		Post							
Variable	M (sd) EG	M (sd) CG	M (sd) EG	M (sd) CG	Interaction *F*(*p*)	η^2^	Group *F*(*p*)	η^2^	Time *F*(*p*)	η^2^
Low threat										
SP-[mm]	263.8(49.43)	259.9(53.15)	217.2(42.11)	257.3(73.45)	7.36(<0.01)	0.27	0.6(>0.05)	0.03	9.23(<0.01)	0.33
SPAP-[mm]	225.1(49.19)	224.2(50.23)	186.2(39.92)	224.7(68.33)	6.15(<0.05)	0.24	0.74(>0.05)	0.04	5.82(<0.05)	0.23
SPML-[mm]	96.7(13.49)	97.73(30.62)	79.7(17.11)	100.6(40.43)	20.49(<0.001)	0.52	0.83(>0.05)	0.04	10.26(<0.01)	0.35
High threat										
SP-[mm]	256.8(61.18)	263.1(65.90)	224.5(69.95)	270.4(78.96)	4.15(>0.05)	0.17	0.82(>0.05)	0.04	1.65(>0.05)	0.08
SPAP-[mm]	219.3(57.86)	222.7(60.54)	179.6(63.29)	230.4(73.45)	6.6(<0.05)	0.25	1.04(>0.05)	0.05	3.0(>0.05)	0.14
SPML-[mm]	95.3(19.49)	99.4(25.17)	85.8(20.20)	100.6(26.46)	3.3(>0.05)	0.14	0.94(>0.05)	0.05	1.99(>0.05)	0.09

Note. HW—height workers; OW—office workers; SP—Sway Path; ML—medio-lateral; AP—antero-posterior; Low threat—measurement at ground level; High threat—measurement at increased height of the measuring platform.

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
