# Peer review of "The Influence of Proprioceptive Training with the Use of Virtual Reality on Postural Stability of Workers Working at Height"

_sensors, 2020, doi:10.3390/s20133731_

Round 1

Reviewer 1 Report

I would like to congratulate the authors on producing an interesting, well presented research paper with meaningful conclusions.

The investigation demonstrates that a virtual reality game intervention, focused on enhancing postural stability, likely does have a significant effect. This can be incorporated into construction worker safety training to help reduce the frequency of height related stability accidents.

The literature survey in the introduction provided sufficient background information on the research field. This is a novel study as no such research has been conducted, however similar literature examples such as using visual/audio/virtual reality feedback interventions on the elderly to assess postural stability improvement are included.

The study was well designed including a reasonable amount of participants, a control group for comparison and multiple training intervention sessions. It was good that the authors included two different virtual reality games with intent on enhancing different aspects related to postural stability.

The methods and technology used were well described. The researchers tested postural stability in a range of different scenarios, highlighting the intervention had an overall positive affect on postural stability.

The only scenario which the intervention did not significantly improve measures associated with postural stability was when the participants had to complete a dual task with eyes closed at a high threat. The authors explained that this could be due to the complexity of the task at hand, especially with eyes being closed.

The authors realise the limitation of their work that it only includes VR technologies related to games on a balance platform. They suggest that VR glasses could be implemented to deepen the immersion and learning process through experiences in VR. It would be good for the authors to provide an example here.

The authors conclude that training on the platform can have positive effects on postural stability that are otherwise difficult to achieve through standard balance exercises. This is reinforced by the results.

Author Response

We would like to thank the reviewer for the constructive and competent criticism. The manuscript has been revised according to the comments raised by the reviewer to the best of our ability. Changes to the manuscript are featured and highlighted in red colour. Please find a detailed reply to the comments attached to this revision.

  1. Comments to the Author

The authors realize the limitation of their work that it only includes VR technologies related to games on a balance platform. They suggest that VR glasses could be implemented to deepen the immersion and learning process through experiences in VR. It would be good for the authors to provide an example here.

Response:

We thank the Reviewer for this important suggestion. The comments and examples have been added in the manuscript, highlighted in red (current line 382-388).

Reviewer 2 Report

The aim of the present study was to evaluate the influences of VR training on postural stability and particularly in workers working at height. In the introduction, the authors well introduced the benefits of VR training and connected balance with working at height. However, the aim was not strong, with no previous research on VR training for balance evaluations in high-altitude workers. Furthermore, there were two factors, proprioceptive training, and VR, conducted in the experiment, which might shade influences on postural stability for each other. Therefore, if the control group did not take any training or exercise, the improvements might result from participants in the experimental group had the proprioceptive training, not necessarily to using the VR device. For the statistical analysis, participants conducted four conditions before and after the intervention. It might not be proper to run eight conditions separately. However, it was a complicated design with four within-subject factors: threat (low and high), task (quiet standing and dual-task), time (pre and post), eyes (open and closed); and one between-subject factor (CG and EG), which is a repeated measured design and should employ repeated measure ANOVA analysis. The authors should provide sufficient explanations for their statistical analysis, which would further affect interpretations and discussions of the outcomes. In the discussion, half of the discussion has discussed the effects of balance platform and VR; however, the balance platform and VR have been confirmed that both could improve balance control individually. The current study, as mentioned above, could not identify balance improvements form either the balance platform or VR due to the CG did not take any training exercise. On the other hand, the authors ran the statistical analysis regarding the factors of threat, task, and eyes separately; then, the explanations might consider discussing the effects of the factors correspondingly, not overstretch the outcomes.

Other comments

  • Methods: any safety prevention was employed to the participants, particularly in the Height-Threat condition?
  • Suggest adding subtitles in the discussion and arrange the paragraphs corresponding to the order of the results or factors.
  • Line 336~338: The comparison between pre and post intervention in the EG could get similar results and provide additional supports for the proprioception training.

Author Response

We would like to thank the Reviewer for the constructive and competent criticism. The manuscript has been revised according to the comments raised by the reviewer to the best of our ability. Changes to the manuscript are featured and highlighted in green colour. Please find a detailed reply to the comments attached to this revision.

  1. Comments to the Author

The aim was not strong, with no previous research on VR training for balance evaluations in high-altitude workers.

Response:

It is true that there are very few study connected to the problem of proprioceptive balance training with use of VR in high-altitude workers. However in our opinion small number of articles in this area is an argument for originality and novelty of our study.

  1. Comments to the Author

There were two factors, proprioceptive training, and VR, conducted in the experiment, which might shade influences on postural stability for each other. Therefore, if the control group did not take any training or exercise, the improvements might result from participants in the experimental group had the proprioceptive training, not necessarily to using the VR device. 

Response:

The Reviewer is right and there are actually two factors acting here. However, it would be difficult in practice to separate this two effects of the device used. The device is based on simultaneous body balance training with the usage of biofeedback associated with VR. The examined person exercises on the platform reacting to changes in the situation provided in the virtual game. On the other hand, we use the term of “proprioceptive training with use VR” in title and aim of the study.

However, the Reviewer's comment is great and we will try to separate these two factors in future studies. It could be taken into account control group with training on platform without VR feedback. Unfortunately, it is impossible in the current experiment.

On the other hand, we used few times in the text “VR training” term what was a  shortcut and might be impropriate. It was corrected in the whole text. But we would like to emphasize that our aim was to check the effect of proprioceptive training with the use of VR but not VR alone.

  1. Comments to the Author

It might not be proper to run eight conditions separately. However, it was a complicated design with four within-subject factors: threat (low and high), task (quiet standing and dual-task), time (pre and post), eyes (open and closed); and one between-subject factor (CG and EG), which is a repeated measured design and should employ repeated measure ANOVA analysis. The authors should provide sufficient explanations for their statistical analysis, which would further affect interpretations and discussions of the outcomes .

Response:

Thank the Reviewer for this remark. We considered such a complex analysis. However in this research we didn’t want to focus on differences between conditions effects. We wanted to explore interaction effect (group x time) for each condition separately. Moreover, analysis in one model can be problematic due to relatively small sample sizes in the context of a large number of comparisons and estimated parameters. The information has been added in the manuscript, highlighted in green (current line 230-232).

  1. Comments to the Author

In the discussion, half of the discussion has discussed the effects of balance platform and VR; however, the balance platform and VR have been confirmed that both could improve balance control individually. The current study, as mentioned above, could not identify balance improvements form either the balance platform or VR due to the CG did not take any training exercise. On the other hand, the authors ran the statistical analysis regarding the factors of threat, task, and eyes separately; then, the explanations might consider discussing the effects of the factors correspondingly, not overstretch the outcomes.

Response:

We thank the Reviewer for this important suggestion. Of course, the Reviewer is right and as we already mentioned there are actually two factors acting here and our response is similar to previous one.

As we stated above, in current experiment is not possible to separate this two factors. So we have added appropriate limitation of the study, highlighted in green (current line 389-392).

  1. Comments to the Author

Methods: any safety prevention was employed to the participants, particularly in the Height-Threat condition?

Answer:

We thank the reviewer for this important remark. The safety prevention has been used as follows:

  • A stable, one-meter-high pedestal was used to place the stabilometry platform at a height. Participants climbed the pedestal via three-step stairs.
  • Gym mattresses were positioned around the platform to ensure safety.

Additional explanation has been added to the section Measurement, highlighted in green (current line 205-207).

  1. Comments to the Author

Suggest adding subtitles in the discussion and arrange the paragraphs corresponding to the order of the results or factors.

Response:

Subtitles in the discussion have been added, highlighted in green (current line 297 and 343).

  1. Comments to the Author

Line 336~338: The comparison between pre and post intervention in the EG could get similar results and provide additional supports for the proprioception training.

Response:

We found significant effects of “time” in most of analyzed variables in most of the conditions. But even in cases with no “time” effect values pre intervention were higher than post intervention. We would like to focus mainly on interaction and main effects – namely, if intervention had significant impact.

Reviewer 3 Report

This manuscripts presents the results of a study exploring the effects of VR training on postural stability of height workers

First introduce the abbreviation in its full form and only than the abbreviated form – virtual reality (VR), experimental group (EG), CG, COP.

Line 18 – Change “eyes open” to “eyes closed”.

Line 20 Change “may support” to “can support”.

Line 62 – What kind of new technology?

Line 65 – Change “According to the research” – “According to research”.

Line 66 – architecture and medicine are not “technologies”.

For a more comprehensive overview, please also add or at least discuss the disadvantages of VR technologies. There are numerous studies showing that often the benefits of virtual reality are surpassed by the limitations due to cyber sickness. Knowing the limitations of a new technology solutions shows a deeper understanding of its overall usefulness. Due to the lack of the “other side” of the story, the Introduction comes out somewhat bias.

Line 115 – “xxx” should be removed.

Table 1. Just to be consistent, add the decimals also for the mean value of age for the EG (34.00)

Line 121 – remove the spacing

Line 147 – at what sampling rate is the data from the gyroscope being sent? As it is real time, what is the delay? Were the timestamps aligned with the biofeedback signals?

Line 173 – just one monitor was used? VR effect is commonly created by VR headsets consisting of a head-mounted display with a small screen in front of the eyes, but can also be created through specially designed rooms with multiple large screens. Please elaborate on how you determined that the used system is a VR system.

Line 197 – what was the cognitive task? Counting backwords from 200?

Line 217 – Did you check for homogeneity of variances and data normality? Which test did you use for that?

Line 291 – I would suggesting rephrasing this sentence. Achieving lower results usually means worse performance. You are trying to report the opposite.

Line 299 – 327 – this part should be moved to the Introduction. The discussion section should focus on interpretation of the results and their application in science or real world applications.

Line 390. Using “I” in the acknowledgment implies that only one author wrote the paper.  I suggest you change it to “we”. If just one of the authors would like to thank Mr. Michal Wejchenig, he or she should use »Name Surname would like to thank…«

Author Response

We would like to thank the Reviewer for the constructive and competent criticism. The manuscript has been revised according to the comments raised by the reviewer to the best of our ability. Changes to the manuscript are featured and highlighted in blue colour. Please find a detailed reply to the comments attached to this revision.

  1. Comments to the Author

First, introduce the abbreviation in its full form and only than the abbreviated form – virtual reality (VR), experimental group (EG), CG, COP.

Line 18 – Change “eyes open” to “eyes closed”.

Line 20  – Change “may support” to “can support”.

Line 62 – What kind of new technology?

Line 65 – Change “According to the research” – “According to research”.

Line 66 – architecture and medicine are not “technologies”.

Response:

We thank the reviewer for this important suggestions. Changes to the manuscript are featured and highlighted in blue as follows.

  • The abbreviated form has been changed as suggested (current line 10-14).
  • Line 18- It has been corrected (current line 18).
  • Line 20- It has been corrected (current line 21).
  • Line 62- Comments has been added (current line 62).
  • Line 65- It has been corrected (current line 65).
  • Line 66- Comments has been added (current line 66-67).
  1. Comments to the Author

For a more comprehensive overview, please also add or at least discuss the disadvantages of VR technologies. There are numerous studies showing that often the benefits of virtual reality are surpassed by the limitations due to cyber sickness. Knowing the limitations of a new technology solutions shows a deeper understanding of its overall usefulness. Due to the lack of the “other side” of the story, the Introduction comes out somewhat bias.

Response:

Additional explanation has been added to the Introduction (current line 84-91).

  1. Comments to the Author
  • Line 115 – “xxx” should be removed.
  • Table 1. Just to be consistent, add the decimals also for the mean value of age for the EG (34.00)
  • Line 121 – remove the spacing.

Response:

  • Line 115- “xxx” has been removed (current line 124).
  • Table 1.- The decimals have been added line
  • The space has been removed.
  1. Comments to the Author

Line 147 – at what sampling rate is the data from the gyroscope being sent? As it is real time, what is the delay? Were the timestamps aligned with the biofeedback signals?

Response:

The sampling rate of the data from the gyroscope is 40Hz/second. The delay is 25ms (due to the transmission frequency of 40/s). The delay resulting from the internal filtering algorithm to ensure the stability of the angle measurement should also be taken into account. The timestamps were aligned with the feedback signals. Addition explanation has been added to the section Intervention, highlighted in blue (current line 153-154).

  1. Comments to the Author

Line 173 – just one monitor was used? VR effect is commonly created by VR headsets consisting of a head-mounted display with a small screen in front of the eyes, but can also be created through specially designed rooms with multiple large screens. Please elaborate on how you determined that the used system is a VR system.

Response:

The Reviewer is right. VR in narrow sense is associated with immersion in digital word and is connected to use the head-mounted display (HMD) or specially designed rooms with multiple large screens.  However, VR is also recognized in wider sense as with use of any digital scenes or exergames. Many times in the articles one can meet the term virtual reality while training on e.g. Nintendo Wii or Play Station. We decided to use the meaning of extended VR just similar like other authors (e.g., Bang, Dong and Kim, 2016; Lee, Kim and Lee, 2015).

As in other VR training, when playing the participants used a wireless system to interact with the avatars on the screen via motion detection system.  The platform was connected by acceleration detecting sensors responding to changes in direction and speed. Due to the detector being installed on the computer, the screen showed the movements of the controller just as the player performed the movements.

  • Bang, Y.S.; Son, K.H.; Kim, H.J. Effects of virtual reality training using Nintendo Wii and treadmill walking exercise on balance and walking for stroke patients. Phys. Ther. Sci., 2016, 28: 3112–3115.
  • Lee, Y.H.; Kim, Y.L.; Lee, S.M. Effects of virtual reality-based training and task-oriented training on balance performance in stroke patients. Phys. Ther. Sci., 2015, 27: 1883–1888.

On the other hand, we wrote about the necessity of providing further study with virtual glasses in the limitation section.

  1. Comments to the Author

Line 197 – what was the cognitive task? Counting backwards from 200?

Response:

Cognitive tasks (i.e., “dual task situations”), we used in our research a mathematical task which consisted of counting every 3 numbers down from 200.

Cognitive tasks are those undertakings that require a person to mentally process new information (i.e., acquire and organize knowledge/learn) and allow them to recall, retrieve that information from memory and to use that information at a later time in the same or similar situation (i.e., transfer).

  1. Comments to the Author

Line 217 – Did you check for homogeneity of variances and data normality? Which test did you use for that?

Response:

We checked normality of data with use of Shapiro-Wilk test. Vast majority of distributions were normal. On the other hand ANOVA is robust for lack of normal distribution.

It was also checked homogeneity of variance with use of Levene test. Test didn’t reach significance at level p<0.05. However, in case of equal sample sizes ANOVA is quite robust for violation of homogeneity.

  1. Comments to the Author

Line 291 – I would suggesting rephrasing this sentence. Achieving lower results usually means worse performance. You are trying to report the opposite.

Response:

Of course, the Reviewer is right, but in this particular case for all the analyzed parameters lower results indicate a higher level of postural stability. The additional explanation has been added to the section Discussion, highlighted in blue (current line 305-306).

  1. Comments to the Author

Line 299 – 327 – this part should be moved to the Introduction. The discussion section should focus on interpretation of the results and their application in science or real world applications.

Response:

We thank the reviewer for these suggestions. In this passage, we have presented and discussed the results of our own research. The results were proof by the results of studies by other authors (Colzato et al. 2013). This is important from the viewpoint of the interpretation of the results.

  1. Comments to the Author

Line 390. Using “I” in the acknowledgment implies that only one author wrote the paper.  I suggest you change it to “we”. If just one of the authors would like to thank Mr. Michal Wejchenig, he or she should use »Name Surname would like to thank…«

Response:

The additional correction has been added to section the Acknowledgments, highlighted in blue, (current line 417).

Round 2

Reviewer 2 Report

No further comments